# Multi-Phase, Contrast-Enhanced Computed Tomography-Based Radiomic Prognostic Marker of Non-Metastatic Pancreatic Ductal Adenocarcinoma

**DOI:** 10.3390/cancers14102476

**Published:** 2022-05-18

**Authors:** Dong Woo Shin, Jaewon Park, Jong-Chan Lee, Jaihwan Kim, Young Hoon Kim, Jin-Hyeok Hwang

**Affiliations:** 1Department of Internal Medicine, Hallym University College of Medicine, Hallym University Sacred Heart Hospital, Anyang 14068, Korea; delight0618@hallym.or.kr; 2Department of Internal Medicine, Seoul National University College of Medicine, Seoul National University Bundang Hospital, Seongnam 13620, Korea; parkjw@scmc.kr (J.P.); ljc0316@snubh.org (J.-C.L.); drjaihwan@snubh.org (J.K.); 3Department of Radiology, Seoul National University College of Medicine, Seoul National University Bundang Hospital, Seongnam 13620, Korea

**Keywords:** radiologic prognostic factor, computed tomography, pancreatic cancer

## Abstract

**Simple Summary:**

Patients with PDA lesions with high attenuation values had longer OS in the RPC and BRPC/LAPC groups and were more likely to undergo a surgical resection after neoadjuvant chemotherapy. These results indicate that intra-tumoral contrast enhancement on CT is an independent prognostic factor in patients with non-metastatic PDA.

**Abstract:**

Background/Aim: This study investigated the predictive ability of intra-tumor enhancement on computed tomography (CT) for the outcomes of patients with pancreatic ductal adenocarcinoma (PDA). Methods: Multi-phase, contrast-enhanced CT (including unenhanced, pancreatic parenchymal phase (PPP) and portal venous phase (PVP)) images of patients diagnosed with non-metastatic PDA were analyzed to investigate prognostic factors. Results: Two hundred ninety-eight patients with PDA (159 with resectable pancreatic cancer (RPC) and 139 with borderline resectable pancreatic cancer (BRPC)/locally advanced pancreatic cancer (LAPC)) were included. The attenuation values of PDA during the PPP (94.5 vs. 60.7 HU; *p* <0.001) and PVP (101.5 vs. 75.5 HU; *p* <0.001) were higher in patients with RPC than in those with BRPC/LAPC. Well-enhanced PDA during the PPP was associated with longer overall survival in the RPC group (27.9 vs. 15.4 months; *p* <0.001) and the BRPC/LAPC group (22.7 vs. 13.6 months; *p* = 0.024). Patients with BRPC/LAPC who underwent neoadjuvant treatment and had well-enhanced PDA during the PPP were more likely to undergo resection. Although tumor size was also an independent prognostic factor, it was not correlated with intra-tumoral enhancement during the PPP. Conclusions: Intra-tumoral contrast enhancement on CT is an independent prognostic factor in patients with non-metastatic PDA.

## 1. Introduction

Pancreatic ductal adenocarcinoma (PDA) is the fourth leading cause of death, and the death rate has increased in the past decade, with an estimated death toll of 25,270 men and 22,950 women in the United States (US) in 2021 [1]. The five-year survival rate of patients with PDA is 9% in the US and 12.2% in South Korea [1,2]. Although complete resection (R0) is a curative treatment, only 15–20% of patients with PDA are surgical candidates [3,4]. Furthermore, disease recurrence occurs in approximately two-thirds of patients even after curative pancreatectomy [5,6]. Given the poor prognosis and high recurrence rate, the identification of prognostic factors is important to determine efficient treatment strategies and predict life expectancy. Several prognostic factors for PDA have been reported including the neutrophil-to-lymphocyte ratio, platelet-to-lymphocyte ratio, tumor-infiltrating platelets, carbohydrate antigen 19-9 (CA19-9), TNM stage, and microRNA biomarkers [7,8,9,10,11,12,13]. However, few studies regarding radiologic markers that can be used to predict the clinical outcome of patients with PDA have been reported.

Multi-phase, contrast-enhanced computed tomography (CT) is an imaging modality that is widely used for diagnosing, staging, and monitoring the treatment responses of patients with PDA [14,15]. PDA is a low-attenuating tumor compared to the adjacent parenchyma during the pancreatic parenchymal and venous phases (PPP and PVP, respectively) of multi-phase CT [16]. The low-attenuating nature of pancreatic cancer may be attributable to intra-tumoral hypoperfusion caused by desmoplasia [17]. While lower tumor vascularity has been associated with biological aggressiveness in tumors [18,19], the prognostic value of tumor enhancement on CT scans in patients with PDA remains unclear [15,20,21]. The purpose of this study was to determine whether intra-tumoral contrast enhancement on CT can be used to predict the clinical outcomes of patients with non-metastatic PDA.

## 2. Materials and Methods

### 2.1. Study Population

Patients who were diagnosed with non-metastatic PDA from April 2009 to July 2017 at Seoul National University Bundang Hospital (SNUBH) were included in this study. This study was approved by the SNUBH Medical Ethics Committee (institutional review board number: B-2102/666-107) on 10 February 2021. The requirement of informed consent was waived by the ethics committee. This study was conducted according to the principles of the Declaration of Helsinki. Patient data were retrospectively retrieved from the Korean Pancreatic Cancer (K-PaC) registry [11,22], and all medical records were anonymized prior to the analyses.

Patients with pathologically confirmed adenocarcinoma who were clinically diagnosed with non-metastatic PDA (resectable pancreatic cancer (RPC), borderline resectable pancreatic cancer (BRPC), or locally advanced pancreatic cancer (LAPC)) by a multidisciplinary tumor board according to the criteria proposed by the National Comprehensive Cancer Network (NCCN) [23] and underwent multi-phase, contrast-enhanced CT (including unenhanced phase (UP), PPP, and PVP) within one-month prior to their pathologic diagnosis at our hospital were included in this study. Patients with a histopathologic diagnosis other than PDA, those who did not undergo multi-phase, contrast-enhanced CT at SNUBH at the time of diagnosis, and those with metastasis at the time of diagnosis were excluded from this study.

### 2.2. CT Imaging Technique

All patients underwent multi-phase contrast-enhanced CT imaging using a pancreas protocol. After the acquisition of non-contrast images, an intravenous contrast material (iohexol (350 mg of iodine per milliliter), Omnipaque; GE Healthcare, United States) was injected via the antecubital vein using a power injector (Stellant D; Medrad, Indianola, PA) at a dose of 1.5 mL/kg and a rate of 3–4 mL/s. CT scans of the PPP and PVP were initiated after the bolus of contrast media 20 and 60 s after an upper abdominal aortic enhancement of 200 Hounsfield units (HU), respectively. Non-contrast and PPP images were acquired from the level of the diaphragm to the level of the umbilicus, and PVP images were obtained from the level of the diaphragm to level of the symphysis pubis. Images were acquired with 64 or 256 multi-detector CT scanners (Brilliance 64, iCT256; Philips Medical Systems, Cleveland, OH, USA). The scanning parameters were as follows: 64 × 0.625 or 128 × 0.6205 mm collimation; a rotation speed of 0.5 s; a pitch of 0.641 or 0.993; and a kvP of 120. The effective mAs ranged from 70 to 390 mAs using an automatic tube current modulation technique (Dose-Right; Philips Medical Systems, Best, the Netherlands). Axial and coronal CT images were reconstructed using filtered back projection with 4 mm-thick sections at 3 mm increments.

### 2.3. Image Analysis

The CT images were reviewed using a picture archiving and communicating system. After selecting the single axial CT image showing the largest cross-sectional area of the PDA tumor, the maximal tumor diameter and intra-tumoral attenuation values were measured by a radiologist with 20 years of experience in abdominal radiology (YHK). This radiologist was aware of the study purpose and patient selection criteria, but was blinded to the clinical and follow-up results. The maximum diameter of the PDA was measured using a straight line. The circle or elliptical regions of interests (ROIs) of the PDA lesion were manually drawn as large as possible. Areas of necrosis, peripancreatic vessels, and artifacts were avoided during measurement, and the ROIs were drawn at the same location for each CT phase (UP, PPP, and PVP).

### 2.4. Statistical Analysis

Continuous variables are presented as mean ± standard deviation (SD) and categorical variables as frequency and proportion. Continuous variables with normal distribution were analyzed using the Student’s *t*-test. The chi-square test was used to analyze differences in categorical variables between the groups. Patients were divided into two groups using the X-tile program (Version 3.6.1, Yale University, New Haven, CT, USA) [24], a bioinformatic tool used to determine optimal cut-off points for survival analysis. The X-tile software tested all possible cut-off points of the target quantitative data using log-rank tests and selected the lowest *p*-value. The survival time was calculated from the date of pathologic diagnosis to the last follow-up date or death. Overall survival (OS) was estimated using the Kaplan–Meier survival curves and compared using the log-rank test. Hazard ratios (HRs) and 95% confidence intervals (95% CIs) were calculated for the univariate model of OS using the Cox proportional hazards model. The effects of different variables (such as sex, serum CA 19-9, tumor location/size, contrast enhancement values (HU) in each phase of CT) on survival were analyzed using a univariate analysis. A multivariate analysis was performed using the Cox proportional hazards regression model to identify relevant factors for survival after treatment for PDA. A two-sided *p*-value of <0.05 was considered statistically significant. The assessment of the change in tumor burden after chemotherapy was based on the revised RECIST guidelines (version 1.1) [25]. All statistical analyses were performed using SPSS, version 25 for Windows (IBM Inc., Armonk, NY, USA) and R, version 3.2.3 (The R Foundation for Statistical Computing, Vienna, Austria; http://www.R-project.org; accessed on 15 August 2021).

## 3. Results

### 3.1. Patient Characteristics

A total of 298 patients diagnosed with non-metastatic PDA were included in this study, including 159 (53.4%) patients with RPC, 41 (13.7%) patients with BRPC, and 98 (32.9%) patients with LAPC. All patients with RPC underwent surgery with no neoadjuvant therapy, and 19 (46.3%) patients with BRPC and 20 (20.4%) patients with LAPC underwent surgery after neoadjuvant treatment (folfirinox 34 [87.2%], gemcitabine 5 [12.8%]). The remaining 100 patients who did not undergo surgery only received palliative chemotherapy (Figure 1).

The median patient age was 64.6 years (range: 30.0–88.4 years), and 171 (57.4%) patients were male (Table 1). The median patient body mass index (BMI) was 22.5 ± 3.0 kg/cm^2^. There were no significant differences in age, sex, or BMI between the RPC and BRPC/LAPC groups. CA19-9 was significantly elevated in the BRPC/LAPC group compared to the RPC group (1157.5 ± 2783.0 vs. 495.2 ± 1046.5 U/mL; *p* = 0.009). PDA was found in the head or neck of the pancreas in 197 (66.1%) patients and in the body or tail of the pancreas in 101 (33.9%) patients. The site of the PDA lesion was not significantly different between the two groups (*p* = 0.070). The mean tumor size was 35.0 ± 13.9 mm (RPC 33.6 ± 13.5 mm vs. BRPC/LAPC 36.6 ± 14.2 mm; *p* = 0.070).

According to the American Joint Committee on Cancer (AJCC) 8th edition, 25 (8.4%) patients had T1 (≤2 cm) tumors, 195 (65.4%) had T2 (2–4 cm) tumors, and 78 (26.2%) had T3 (>4 cm) tumors. In the unenhanced images, the measured attenuation of the PDA tumor was not significantly different between the RPC and BRPC/LAPC groups (37.3 ± 6.9 HU vs. 35.8 ± 7.7 HU; *p* = 0.078). However, the attenuation values of the PDA tumor were significantly higher in the RPC group than in the BRPC/LAPC group in both the PPP (94.5 ± 27.5 HU vs. 60.7 ± 19.6 HU; *p* < 0.001) and the PVP (101.5 ±/27.5 HU vs. 75.5 ± 25.9 HU; *p* < 0.001) (Figure 2).

The postoperative pathology results of the 198 patients who underwent surgical resection are as follows: tumor differentiation (well-differentiated 21 (10.6%), moderately differentiated 160 (80.8%), poorly differentiated 17 (8.6%)), pT-stage (pT1 22 (11.1%), pT2 131 (66.2%), pT3 45 (22.7%)), metastatic lymph node status (pN0 81 (40.9%), pN1 78 (39.4%), pN2 39 (19.7%)), resection margin (R0 155 (78.3%), R1 43 (21.7%)), lymphovascular invasion (LVI− 78 (39.4%), LVI+ 120 (60.6%)), and perineural invasion (PNI− 24 (12.1%), PNI+ 174 (87.9%)). Of the 198 patients who underwent surgical resection, 123 (62.1%) received adjuvant chemotherapy (gemcitabine 103 (52.0%), 5-fluorouracil 20 (10.1%)).

### 3.2. Survival Analysis

The median OS was 18.2 months (95% CI: 15.8–20.6 months), with a median follow-up period of 19.5 months (range: 1.1–82.2 months). The median OS of patients was 22.0 months with RPC (95% CI: 17.8–26.2 months), 16.0 months with BRPC (95% CI: 12.2–19.8 months), and 14.0 months with LAPC (95% CI: 11.2–16.7 months) (*p* = 0.002).

#### 3.2.1. OS in Patients with RPC

Patients with RPC who had an intra-tumoral enhancement ≥92.8 HU during the PPP had a significantly longer median OS than those with an intra-tumoral enhancement <92.8 HU (27.9 vs. 15.4 months; *p* <0.001) (Table 2 and Figure 3A). Patients with RPC who had an intra-tumoral enhancement ≥99.8 HU during the PVP had a significantly longer median OS than those with an intra-tumoral enhancement <99.8 HU (25.5 vs. 15.4 months; *p* <0.001) (Table 2). The T-stage according to tumor size (T2, HR = 2.254, 95% CI = 1.033–4.919, *p* = 0.041; T3, HR = 4.955, 95% CI = 2.157–11.383, *p* <0.001) and intra-tumoral attenuation values during the PPP (≥92.8 HU, HR = 0.445, 95% CI = 0.301–0.657, *p* <0.001) and PVP (≥99.8 HU, HR = 0.500, 95% CI = 0.339–0.738, *p* <0.001) were significantly associated with OS. Sex (*p* = 0.350), CA19-9 (*p* = 0.748), tumor location (*p* = 0.606), and attenuation values in UP (*p* = 0.899) were not significantly associated with OS. T3 (>4 cm, adjusted HR (aHR) = 4.050, 95% CI = 1.750–9.376, *p* <0.001) and enhancement during the PPP (≥92.8 HU, aHR = 0.487, 95% CI = 0.328–0.722, *p* <0.001) were identified as independent prognostic factors for OS (Table 3).

#### 3.2.2. OS in Patients with BRPC and LAPC

Patients with BRPC/LAPC with intra-tumoral enhancement ≥84.9 HU during the PPP had a significantly longer median OS than those with intra-tumoral enhancement <84.9 HU (22.7 vs. 13.6 months; *p* = 0.024) (Table 2 and Figure 3B). Patients with BRCP/LAPC with an intra-tumoral enhancement ≥101.0 HU during the PVP had a significantly longer median OS than those with an intra-tumoral enhancement <101.0 HU (21.6 vs. 13.6 months; *p* = 0.050). T3 (>4 cm, aHR = 3.335, 95% CI = 1.007–11.039, *p* = 0.049) and attenuation values during the PPP (≥84.9 HU, aHR = 0.497, 95% CI = 0.226–0.950, *p* = 0.009) were identified as prognostic factors for OS (Table 4).

The median OS was significantly longer in patients with well-enhanced (≥48.6 HU) PDA than those with poorly enhanced (<48.6 HU) PDA (16.1 vs. 8.9 months; *p* = 0.007) during the PPP of the follow-up multi-phase CT obtained after chemotherapy. T3 (aHR = 2.825, 95% CI = 1.580–5.051, *p* <0.001) and attenuation during the PPP (≥48.6 HU, aHR = 0.767, 95% CI = 0.479–0.952, *p* = 0.009) were independent prognostic factors for OS (see Appendix A, which shows the factors associated with prognosis in patients with BRPC/LAPC). Patients who underwent a pancreatectomy after neoadjuvant chemotherapy (*n* = 39) had a longer OS than patients treated with palliative chemotherapy alone (*n* = 100) (30.0 vs. 12.1 months; *p* <0.001). The median OS and resection rates were longer and higher in patients with increased enhancement during the PPP on follow-up CT (median OS, 16.1 vs. 13.0 months; surgical resection rate, 33.8 vs. 21.0%) and in those with reduced tumor size on follow-up CT (median OS, 17.3 vs. 10.8 months; surgical resection rate, 39.3 vs. 10.9%).

Among 139 patients in the BRPC/LAPC group, CA19-9 decreased in 93 patients (66.9%) and increased in 46 patients (33.1%) after neoadjuvant chemotherapy. Patients with decreased CA19-9 had a significantly longer survival period than those with elevated CA19-9 (18.6 months (95% CI = 15.1–22.1) vs. 11.2 months (95% CI = 10.1–12.4); *p* < 0.001).

### 3.3. Correlation between T-Stage and Intra-Tumoral Enhancement

Although relatively large tumors showed poor contrast enhancement in both groups, the enhancement was not significantly different (see Appendix A, which shows the relationship between tumor size and contrast enhancement). The Pearson correlation coefficient for intra-tumor enhancement during the PPP was −0.261. Tumor size and contrast enhancement were found to be independent prognostic factors.

## 4. Discussion

A CT-based radiographic biomarker that can be used to predict the prognosis of patients with non-metastatic PDA was identified in this study. The results of this study show that the intra-tumoral attenuation values in contrast-enhanced CT were higher in patients with RPC than in those with BRPC/LAPC. Patients with PDA lesions with high attenuation values had longer OS in the RPC and BRPC/LAPC groups and were more likely to undergo a surgical resection after neoadjuvant chemotherapy. These results indicate that intra-tumoral contrast enhancement on CT is an independent prognostic factor in patients with non-metastatic PDA.

According to the NCCN guidelines, multi-phase, contrast-enhanced CT is the preferred initial imaging modality to achieve an accurate diagnosis and determine the appropriate management strategies for patients with PDA. Most PDA lesions are low-attenuating tumors during the PPP and low- to iso-attenuating tumors during the PVP [16,20]. However, approximately 10% of PDA lesions are iso-attenuating compared to the pancreatic parenchyma, rendering the diagnosis of PDA difficult [26,27]. Tissues with abundant arterial blood supply generally have their highest enhancement in the arterial phase. In contrast, desmoplasia disturbs the contrast inflow to the tumor, resulting in poor enhancement in the early phase. Therefore, PDA lesions typically exhibit a progressively delayed enhancement pattern as a reflection of the dense fibrotic deposition within and adjacent to the tumor [20,28]. Hypoxia caused by poor vascularity is an important factor for the development of aggressive PDA and is accepted as a major contributor to resistance against chemotherapy [29,30]. Therefore, reduced intra-tumoral enhancement may be useful for the prediction of treatment resistance and poorer long-term outcomes in patients with PDA [15]. Based on these findings, this study investigated whether contrast enhancement in multi-phase CT can be a significant radiologic prognostic factor in patients with PDA.

In this study, patients with well-enhanced PDA lesions during the PPP had longer OS than those with poorly enhanced PDA in the RPC and BRPC/LAPC groups. In addition, patients with RPC had higher contrast enhancement than patients with unresectable PDA. Previous studies have reported that iso-attenuating PDA lesions are associated with a longer OS than hypo-attenuating PDA lesions [20,26,27]. Yoon et al. reported that poorly detected, small PDA lesions that were iso-attenuating on the initial CT scan had an increased size and decreased attenuation after six months, which is consistent with the results of this study that patients with RPC have higher-attenuating lesions than those with BRPC/LAPC [20]. In addition, the prevalence of iso-attenuating PDA lesions was significantly higher among patients with small (<20 mm) or well-differentiated tumors than among those with moderately or poorly differentiated tumors [20]. Fukukura et al. reported that contrast enhancement during the PPP was the most significant predictor of survival in patients with PDA [31]. Decreased tumor attenuation on enhanced CT may reflect changes in tumor composition, indicating that tumor cells live in a harsh tumor microenvironment (TME) as the tumor grows. The TME regulates essential tumor survival and promotion functions. Interactions between the cellular and structural components of the TME allow cancer cells to become invasive and disseminate from the primary site to distant locations through a complex and multistep metastatic cascade [32].

The tumoral attenuation values obtained during the PPP were not closely correlated with tumor size (T-stage) and were an independent prognostic marker in patients in the RPC and BRPC/LAPC groups. T-stage classified by tumor size has been reported as a significant predictor of survival [11,12]. Rapidly growing PDA lesions frequently undergo hypoxia-induced necrosis due to an insufficient blood supply. Tumor necrosis, as an important prognostic histological parameter in patients with PDA, has been previously reported [33,34]. In many types of cancer, tumor necrosis is a result of hypoxia, and it has been reported to accelerate the potential for malignancy [29,35]. Therefore, CT attenuation may indicate tumor growth rate in addition to tumor size.

This study has several limitations. First, as this was a retrospective study, the parameters reflecting contrast enhancement were not completely controlled. Histopathologic factors (such as tumor differentiation, cellularity, pattern of intra-tumoral stroma, and necrosis) and CT scanning parameters (including injection timing, the rate of contrast agents, body weight, and cardiac output) can influence tissue enhancement on CT [20,27,36]. Therefore, it is difficult to use the cutoff values of this study as absolute values for clinical applications. Second, contrast enhancement may be interpreted subjectively, as standardized criteria are lacking. In this study, the imaging analysis was performed by a single radiologist; therefore, interobserver reproducibility was not validated for the measurement of the ROIs. Further external validation using a uniform CT protocol in a prospective study is needed. Despite these limitations, this study has notable advantages. First, this study has a larger study population compared to similar studies. Second, radiologic indicators that can predict prognosis enable individualized treatment planning. With more certainty based on radiologic prognostic factors, aggressive chemotherapy can be administered to patients who are expected to have a better prognosis. Further development of imaging-based quantitative biomarkers can be helpful in the clinical decision-making process in patients with PDA.

## 5. Conclusions

In conclusion, patients with poorly enhanced PDA lesions had significantly shorter OS than those with well-enhanced PDA lesions, suggesting that contrast-enhancement on CT is an important prognostic factor in patients with PDA, regardless of treatment methods.

## Figures and Tables

**Figure 1 cancers-14-02476-f001:**
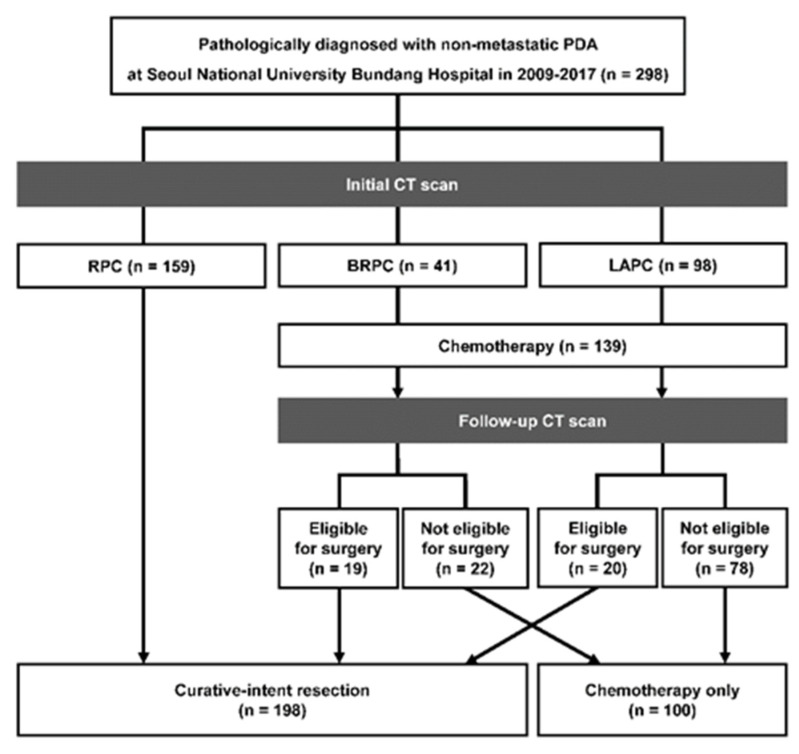
Flowchart of the study population.

**Figure 2 cancers-14-02476-f002:**
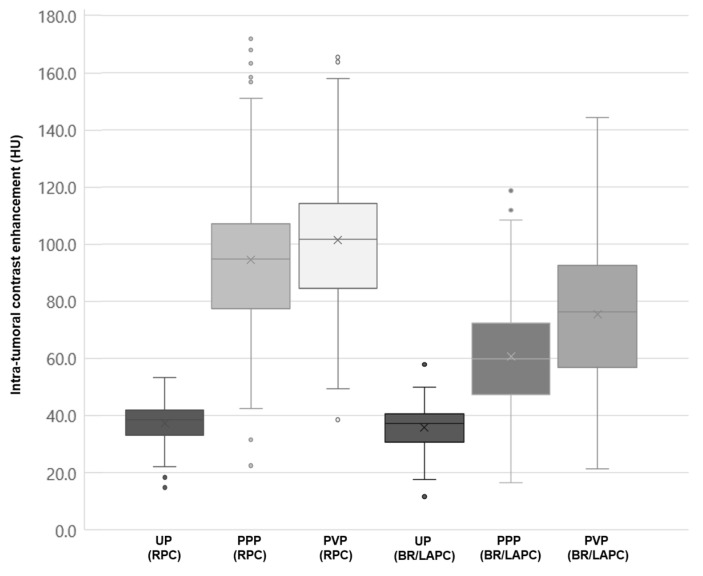
Intra-tumoral contrast enhancement in each phase. (x: median, dots: outliers).

**Figure 3 cancers-14-02476-f003:**
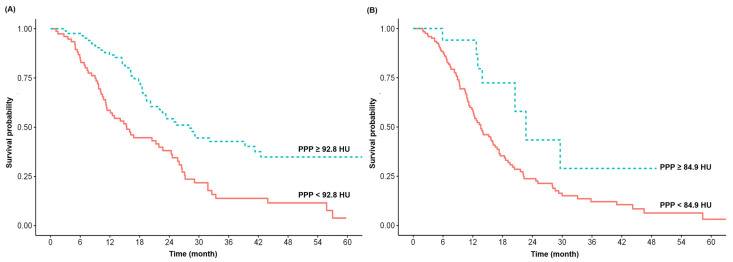
Kaplan–Meier curves showing the relationship between radiologic parameters and overall survival. (**A**) Contrast enhancement during the pancreatic parenchymal phase (PPP) and overall survival (OS) in patients with resectable pancreatic cancer. (**B**) Contrast enhancement during the PPP and OS in patients with borderline resectable or locally advanced pancreatic cancer.

**Table 1 cancers-14-02476-t001:** Baseline characteristics of the study population (*n* = 298).

	RPC (*n* = 159)	BRPC/LAPC (*n* = 139)	Total (*n* = 298)	*p*-Value
Age (years) *	65.8 ± 10.9	63.2 ± 10.6	64.6 ± 10.8	0.390
Male	93 (58.5%)	78 (56.1%)	171 (57.4%)	0.767
BMI (kg/cm^2^) *	22.6 ± 3.2	22.3 ± 2.7	22.5 ± 3.0	0.326
CA 19-9 (U/mL)	495.2 ± 1046.5	1157.5 ± 2783.0	804.1 ± 2071.4	0.009
Tumor location, *n* (%)				
Head or neck	113 (71.1%)	84 (60.4%)	197 (66.1%)	0.070
Body or tail	46 (28.9%)	55 (39.6%)	101 (33.9%)	
Tumor size (mm) *	33.6 ± 13.5	36.6 ± 14.2	35.0 ± 13.9	0.070
T-stage, *n* (%)				
T1 (≤2 cm)	17 (10.7%)	8 (5.8%)	25 (8.4%)	0.039
T2 (2–4 cm)	109 (68.6%)	86 (61.9%)	195 (65.4%)	
T3 (>4 cm)	33 (20.8%)	45 (32.4%)	78 (26.2%)	
Intra-tumoral attenuation values				
UP (HU) *^#^	37.3 ± 6.9	35.8 ± 7.7	36.6 ± 7.3	0.078
PPP (HU) *^#^	94.5 ± 27.5	60.7 ± 19.6	78.8 ± 29.4	<0.001
PVP (HU) *^#^	101.5 ± 27.5	75.5 ± 25.9	89.4 ± 29.7	<0.001

Values are numbers of patients, with percentages in parentheses, unless otherwise specified. * Values are mean ± standard deviation. ^#^ Hounsfield units (HU), representing radiograph attenuation. Abbreviations: RPC, resectable pancreatic cancer; BRPC, borderline resectable pancreatic cancer; LAPC, locally advanced pancreatic cancer; BMI, body mass index; CA19-9, carbohydrate antigen 19-9; HU, Hounsfield unit; UP, unenhanced phase; PPP, pancreatic parenchymal phase; PVP, portal venous phase.

**Table 2 cancers-14-02476-t002:** Median overall survival according to subgroup.

	RPC (*n* = 159)				BRPC/LAPC (*n* = 139)			
	Subgroups	Patients (%)	Median OS (95% CI) (mo.)	*p*-Value	Subgroups	Patients (%)	Median OS (95% CI) (mo.)	*p*-Value
Initial CT								
PPP	<92.8 HU	76 (47.8%)	15.4 (11.0–19.8)	<0.001	<84.9 HU	122 (87.8%)	13.6 (10.9–16.4)	0.024
	≥92.8 HU	83 (52.2%)	27.9 (21.7–34.0)		≥84.9 HU	17 (12.2%)	22.7 (17.6–27.8)	
PVP	<99.8 HU	58 (36.5%)	15.4 (10.0–20.8)	<0.001	<101.0 HU	119 (85.6%)	13.6 (10.8–16.4)	0.050
	≥99.8 HU	101 (63.5%)	25.5 (20.0–31.0)		≥101.0 HU	20 (14.4%)	21.6 (18.4–24.8)	
Follow-up CT								
PPP	NA	NA	NA	NA	<48.6 HU	35 (25.2%)	8.9 (5.0–12.9)	0.007
	NA	NA	NA	NA	≥48.6 HU	104 (74.8%)	16.1 (12.9–19.3)	
PVP	NA	NA	NA	NA	<52.0 HU	22 (15.8%)	6.8 (4.0–9.7)	<0.001
	NA	NA	NA	NA	≥52.0 HU	117 (84.2%)	16.1 (12.4–19.8)	

Abbreviations: RPC, resectable pancreatic cancer; BRPC, borderline resectable pancreatic cancer; LAPC, locally advanced pancreatic cancer; CT, computed tomography; PPP, pancreatic parenchymal phase; PVP, portal venous phase; HU, Hounsfield unit; OS, overall survival; CI, confidence interval; NA, not applicable.

**Table 3 cancers-14-02476-t003:** Univariate and multivariate Cox hazard regression analysis of clinical and radiologic parameters in patients with resectable pancreatic cancer on initial CT scan (*n* = 159).

	Subgroup	Patients (%)	Median OS (95% CI) (Months)	*p*-Value	Univariate Analysis		Multivariate Analysis	
					HR (95% CI)	*p*-Value	aHR (95% CI)	*p*-Value
Sex	Male	93 (58.5%)	20.5 (16.8–24.2)	0.349	1 (reference)	0.350	-	-
	Female	66 (41.5%)	23.5 (13.9–33.0)		0.829 (0.560–1.228)		-	-
CA 19-9	<37 U/mL	40 (25.2%)	23.3 (8.4–38.3))	0.260	1 (reference)	0.748	-	-
	≥37 U/mL	119 (74.8%)	21.9 (17.5–26.3)		1.300 (0.823–2.053)		-	-
Tumor location	Head, neck	113 (71.1%)	22.6 (18.0–27.2)	0.606	1 (reference)	0.606	-	-
	Body, tail	46 (28.9%)	21.3 (11.8–30.8)		1.117 (0.733–1.701)		-	-
Tumor size	T1 (≤ 2 cm)	17 (10.7%)	49.3 (35.7–62.9)	<0.001	1 (reference)		1 (reference)	
	T2 (2–4 cm)	109 (68.6%)	22.7 (17.9–27.5)		2.254 (1.033–4.919)	0.041	1.870 (0.851–4.112)	0.119
	T3 (>4 cm)	33 (20.8%)	14.4 (9.7–19.1)		4.955 (2.157–11.383)	<0.001	4.050 (1.750–9.376)	<0.001
UP	<29.1 HU	16 (10.1%)	16.9 (5.1–28.8)	0.116	1 (reference)	0.899	-	-
	≥29.1 HU	143 (89.9%)	22.7 (18.2–27.2)		0.646 (0.373–1.119)		-	-
PPP	<92.8 HU	76 (47.8%)	15.4 (11.0–19.8)	<0.001	1 (reference)	<0.001	1 (reference)	<0.001
	≥92.8 HU	83 (52.2%)	27.9 (21.7–34.0)		0.445 (0.301–0.657)		0.487 (0.328–0.722)	
PVP	<99.8 HU	58 (36.5%)	15.4 (10.0–20.8)	<0.001	1 (reference)	<0.001	1 (reference)	0.918
	≥99.8 HU	101 (63.5%)	25.5 (20.0–31.0)		0.500 (0.339–0.738)		0.970 (0.539–1.745)	

Abbreviations: RPC, resectable pancreatic cancer; OS, overall survival; HR, hazard ratio; aHR, adjusted hazard ratio; CI, confidence interval; CA19-9, carbohydrate antigen 19-9; UP, unenhanced phase; PPP, pancreatic parenchymal phase; PVP, portal venous phase; HU: Hounsfield unit.

**Table 4 cancers-14-02476-t004:** Univariate and multivariate Cox hazard regression analysis of clinical and radiologic parameters in patients with borderline resectable or locally advanced pancreatic cancer on initial CT scan (*n* = 139).

	Subgroup	Patients (%)	Median OS (95% CI) (mo.)	*p*-Value	Univariate Analysis		Multivariate Analysis	
					HR (95% CI)	*p*-Value	aHR (95% CI)	*p*-Value
Sex	Male	78 (56.1%)	13.6 (10.6–16.7)	0.197	1 (reference)	0.199		
	Female	61 (43.9%)	15.7 (12.4–19.1)		0.768 (0.513–1.149)			
CA 19-9	<37 U/mL	30 (21.6%)	21.5 (10.5–32.6)	0.280	1 (reference)	0.282		
	≥37 U/mL	109 (78.4%)	13.8 (11.2–16.3)		1.301 (0.806–2.100)			
Tumor location	Head, neck	84 (60.4%)	13.6 (10.5–16.8)	0.344	1 (reference)	0.345		
	Body, tail	55 (39.6%)	16.1 (12.5–19.7)		0.822 (0.547–1.235)			
Tumor size	T1 (≤ 2 cm)	8 (5.8%)	42.3 (20.1–64.5)	0.009	1 (reference)		1 (reference)	
	T2 (2–4 cm)	86 (61.9%)	16.5 (14.5–18.5)		2.138 (0.658–6.945)	0.206	2.198 (0.676–7.146)	0.190
	T3 (>4 cm)	45 (32.4%)	11.4 (7.9–14.8)		3.633 (1.100–12.000)	0.034	3.335 (1.007–11.039)	0.049
UP	<30.5 HU	31 (22.3%)	12.3 (9.5–15.1)	0.346	1 (reference)	0.347		
	≥30.5 HU	108 (77.7%)	15.2 (12.9–17.4)		0.802 (0.506–1.271)			
PPP	<84.9 HU	122 (87.8%)	13.6 (10.9–16.4)	0.024	1 (reference)	0.029	1 (reference)	0.009
	≥84.9 HU	17 (12.2%)	22.7 (17.6–27.8)		0.425 (0.197–0.916)		0.497 (0.226–0.950)	
PVP	<101.0 HU	119 (85.6%)	13.6 (10.8–16.4)	0.050	1 (reference)	0.054		
	≥101.0 HU	20 (14.4%)	21.6 (18.4–24.8)		0.540 (0.288–1.011)			

Abbreviations: BRPC, borderline resectable pancreatic cancer; LAPC, locally advanced pancreatic cancer; OS, overall survival; HR, hazard ratio; aHR, adjusted hazard ratio; CI, confidence interval; CA19-9, carbohydrate antigen 19-9; UP, unenhanced phase; PPP, pancreatic parenchymal phase; PVP, portal venous phase; HU: Hounsfield unit.

## Data Availability

The data presented in this study are available on request from the corresponding author.

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
