# Peer review of "Multi-Phase, Contrast-Enhanced Computed Tomography-Based Radiomic Prognostic Marker of Non-Metastatic Pancreatic Ductal Adenocarcinoma"

_cancers, 2022, doi:10.3390/cancers14102476_

Round 1

Reviewer 1 Report

We thank the authors for this beautiful work, here are some comments.

Page 2  Ligne 55-56 : "Intra-tumoral hypoperfusion and hypoxia caused by 55
a dense fibro-inflammatory microenvironment with a desmoplastic reaction may con tribute to the tumor’s low attenuation [17]"

This reference is not clear from the sentence, can you find a reference that illustrates your argument or rephrase it?

page 2 ligne 58 : ", "the prognostic value of tumor  enhancement on CT scans in patients with PDA remains unclear."

References are missing to illustrate this point as there are several published data 

Page 3 Ligne 102 " After selecting the single axial CT image showing the largest cross-sectional area of 101 the PDA tumor, the maximal tumor diameter and intra-tumoral attenuation values were measured by a radiologist with 20 years of experience in abdominal radiology (Y.H.K.). This radiologist was aware of the study purpose and patient selection criteria, but was  blinded to the clinical and follow-up results. "

Has an external centralized review of the images been carried out? 

Author Response

RE: cancers-1696761, entitled "A Multi-phase, Contrast-enhanced Computed Tomography-Based Radiomic Prognostic Marker of Non-metastatic Pancreatic Ductal Adenocarcinoma" Coauthored by Dong Woo Shin, Jaewon Park, Jong-chan Lee, Jaihwan Kim, Young Hoon Kim, Jin-Hyeok Hwang

Thank you very much for giving us an opportunity for revision.

Accurate and kind comments by the reviewer have been addressed in the discussion. We also believe that these comments improved our manuscript. Changes have been made by changing the color to Red in the revised manuscript to avoid any confusion.

I anticipate good response.

Thank you!

Sincerely,

Jin-Hyeok Hwang, M.D., Ph.D.

Dong Woo Shin, M.D.

Reply to Reviewer’s comments

We thank the authors for this beautiful work, here are some comments.

Question #1: Page 2 Line 55-56: "Intra-tumoral hypoperfusion and hypoxia caused by a dense fibro-inflammatory microenvironment with a desmoplastic reaction may contribute to the tumor’s low attenuation [17]" This reference is not clear from the sentences, can you find a reference that illustrates your argument or rephrase it?

Author’s comments: Thank you for your kind comment. We replace this sentence and reference with the following (page 2, line 56-57): “Low-attenuating nature of pancreatic cancer may be attributable to intra-tumoral hypoperfusion caused by desmoplasia [17].”

Question #2: page 2 line 58: ", "the prognostic value of tumor enhancement on CT scans in patients with PDA remains unclear." References are missing to illustrate this point as there are several published data.

Author’s comments: Thank you for your important suggestion. We add references as follows (page 2, line 59-60): “While lower tumor vascularity has been associated with biological aggressiveness in tumors [18,19], the prognostic value of tumor enhancement on CT scans in patients with PDA remains unclear [15,20,21].”

Question #3: Page 3 Line 102 " After selecting the single axial CT image showing the largest cross-sectional area of 101 the PDA tumor, the maximal tumor diameter and intra-tumoral attenuation values were measured by a radiologist with 20 years of experience in abdominal radiology (Y.H.K.). This radiologist was aware of the study purpose and patient selection criteria but was blinded to the clinical and follow-up results." Has an external centralized review of the images been carried out?

Author’s comments: Thank you for your important comment. Unfortunately, external validation was not performed in this study. Therefore, the limitation is described as follows (page 10, line 315-318): “In this study, the imaging analysis was performed by a single radiologist; therefore, interobserver reproducibility was not validated for the measurement of the ROIs. Further external validation using a uniform CT protocol in a prospective study is needed.”

The authors really appreciated the reviewer’s kind and accurate comments. The revision based on these comments made this manuscript more accurate and the quality improved. Thank you again.

Jin-Hyeok Hwang, M.D., Ph.D.

Dong Woo Shin, M.D.

Reviewer 2 Report

Dear Authors:

The authors have carried out the a study tittled “A Multi-phase, Contrast-enhanced Computed Tomography Based Radiomic Prognostic Marker of Non-metastatic Pancreatic Ductal Adenocarcinomace”. This retrospective study analized included 298 patients diagnosed with non-metastatic  pancreatic ductal adenocarcinoma  in order to determine whether intra-tumoral contrast enhancement on CT can be used  to predict the clinical outcomes of this type of patients.

It  is a well designed and elaborated study to investigate the predictive ability of intra-tumor enhancement on computed tomography for the outcomes of patients with a disease with a dismal prognosis.

Some considerations need to be taken into account:

  • Please review table 3, the section corresponding to the gender variable (25.2%) has a couple of meaningless symbols
  • Authors should put more emphasis in the discussion on the reason why the parameters reflecting contrast enhancement were not completely controlled in the study because it can lead to confusion and reduce the power of the study.
  • The absence of a validated protocol for the interpretation of contrast enhancement and the high possibility of subjective interpretations between observers confers a special risk of different biases in the interpretation that could affect the results and the obtaining of coherent and extrapolable conclusions.
  • The evaluation of the study by a single radiologist suggests that the results are observer dependent and subject to a single interpretation with no possibility of validation.
  • As a curiosity, it is noteworthy that the study has been developed in a different center from the work center of the first author of the manuscript.
  • Bibliographical references should be homogenized. Ref num 5 does not have its DOI added. Ref num 9 and 30 are incomplete and not up-dated. Must be correctly cited.

Kind regards

Author Response

RE: cancers-1696761, entitled "A Multi-phase, Contrast-enhanced Computed Tomography-Based Radiomic Prognostic Marker of Non-metastatic Pancreatic Ductal Adenocarcinoma"

Coauthored by Dong Woo Shin, Jaewon Park, Jong-chan Lee, Jaihwan Kim, Young Hoon Kim, Jin-Hyeok Hwang

Thank you very much for giving us an opportunity for revision.
Accurate and kind comments by the reviewer have been addressed in the discussion. We also believe that these comments improved our manuscript. Changes have been made by changing the color to Red in the revised manuscript to avoid any confusion. 

I anticipate good response.
Thank you!
Sincerely,
Jin-Hyeok Hwang, M.D., Ph.D.
Dong Woo Shin, M.D.

Reply to Reviewer’s comments
Dear Authors:
The authors have carried out a study titled “A Multi-phase, Contrast-enhanced Computed Tomography Based Radiomic Prognostic Marker of Non-metastatic Pancreatic Ductal Adenocarcinoma”. This retrospective study analyzed included 298 patients diagnosed with non-metastatic pancreatic ductal adenocarcinoma in order to determine whether intra-tumoral contrast enhancement on CT can be used to predict the clinical outcomes of this type of patients.
It is a well-designed and elaborated study to investigate the predictive ability of intra-tumor enhancement on computed tomography for the outcomes of patients with a disease with a dismal prognosis. Some considerations need to be taken into account:

Question #1: Please review table 3, the section corresponding to the gender variable (25.2%) has a couple of meaningless symbols

Author’s Comments: Thank you for the thorough review. The corresponding part in Table 3 has been corrected.

Authors should put more emphasis in the discussion on the reason why the parameters reflecting contrast enhancement were not completely controlled in the study because it can lead to confusion and reduce the power of the study.

Author’s Comments: Thank you for your important comment. Because it is a retrospective study, the protocol for CT imaging could not be completely controlled. Because various factors can affect vascular enhancement on CT, these limitations were described as follows in the discussion (page 9-10, line 308-313). “First, as this was a retrospective study, the parameters reflecting contrast enhancement were not completely controlled. Histopathologic factors (such as tumor differentiation, cellularity, pattern of intra-tumoral stroma, and necrosis) and CT scanning parameters (including injection timing, the rate of contrast agents, body weight, and cardiac output) can influence tissue enhancement on CT [20,27,36].”

The absence of a validated protocol for the interpretation of contrast enhancement and the high possibility of subjective interpretations between observers confers a special risk of different biases in the interpretation that could affect the results and the obtaining of coherent and extrapolable conclusions. The evaluation of the study by a single radiologist suggests that the results are observer dependent and subject to a single interpretation with no possibility of validation.

Author’s comments: Thank you for your important comment. Unfortunately, external validation was not performed in this study. Therefore, the limitation is described as follows (page 10, line 315-318): “In this study, the imaging analysis was performed by a single radiologist; therefore, interobserver reproducibility was not validated for the measurement of the ROIs. Further external validation using a uniform CT protocol in a prospective study is needed.”

As a curiosity, it is noteworthy that the study has been developed in a different center from the work center of the first author of the manuscript.

Author’s Comments: Dong Woo Shin, the first author of this study, is now working as a faculty at Hallym University Sacred Heart Hospital after completing a fellowship program at Seoul National University Bundang Hospital. Therefore, the affilation was modified as follows (page 1, line 5): “Dong Woo Shin1 → Dong Woo Shin1,2”

Bibliographical references should be homogenized. Ref num 5 does not have its DOI added. Ref num 9 and 30 are incomplete and not updated. Must be correctly cited.

Author’s Comments: Thank you for your comment. We fixed the reference as the reviewer said.

The authors really appreciated the reviewer’s kind and accurate comments. The revision based on these comments made this manuscript more accurate and the quality improved. Thank you again. 

Jin-Hyeok Hwang, M.D., Ph.D.
Dong Woo Shin, M.D.

Reviewer 3 Report

Dr Shin et al. have submitted an interesting retrospective study with the aim to assess the potential preoperative prognostic role of Contrast-enhanced CT scan. Currently, PDAC management is mainly based on radiological resectability criteria and biological criteria, such as CA 19.9 level (when available); definition of new preoperative prognostic markers, able to improve decision-making process (i.e. surgery upfront vs neoadjuvant therapy), is more and more necessary.

This is a good study with a consistent sample size; however, some issues should be addressed to improve the manuscript.

    • Table 1 reported characteristics of study population: no data are reported about tumor grading, lymph nodal status, resection margin status (R0-R1), vascular resections, neoadjuvant regimens, adjuvant chemotherapy (which regimen? Administered or not?). These are well-known prognostic parameters that should be reported and investigated in order to define the real weight of CT attenuation on prognosis.
    • Preoperative CA 19.9 is a well-studied parameter and values > 500 kU/L are considered expression of aggressive tumor and defined as “biological” borderline-resectable disease, according to International Study Group of Pancreas Surgery (ISGPS). Authors should report at least mean or median value of analyzed subgroups and assess its prognostic weight, instead of categorization in “normal/elevated”.
    • Decrease of CA 19.9 is considered a relevant response parameter to neoadjuvant chemotherapy: could authors add data about it in BR/LA PDAC subgroup?
    • How were CT attenuation cut-off, reported in tab. 3 and 4, defined? Could pancreas texture affect tumour CT attenuation (i.e. “fatty pancreas, chronic pancreatitis, etc)? Please clarify.
    • In discussion section (lines 270-271): authors stated that prevalence of iso-attenuating PDAC lesions was higher in small or well-differentiated lesions but no data about grading are reported. Authors should add data about it in results section.
    • Table S2: there is strong difference between pre- and post-neoadjuvant chemotherapy PVP attenuation in T1 BR/LA cancers. Is this difference significant? Could it be considered as better response of small lesions to chemotherapy? This is an interesting finding: authors should analyze it in discussion section.

Author Response

RE: cancers-1696761, entitled "A Multi-phase, Contrast-enhanced Computed Tomography-Based Radiomic Prognostic Marker of Non-metastatic Pancreatic Ductal Adenocarcinoma"

Coauthored by Dong Woo Shin, Jaewon Park, Jong-chan Lee, Jaihwan Kim, Young Hoon Kim, Jin-Hyeok Hwang

Thank you very much for giving us an opportunity for revision.

Accurate and kind comments by the reviewer have been addressed in the discussion. We also believe that these comments improved our manuscript. Changes have been made by changing the color to Red in the revised manuscript to avoid any confusion.

I anticipate good response.

Thank you!

Sincerely,

Jin-Hyeok Hwang, M.D., Ph.D.

Dong Woo Shin, M.D.

Reply to Reviewer’s comments

Dr Shin et al. have submitted an interesting retrospective study with the aim to assess the potential preoperative prognostic role of Contrast-enhanced CT scan. Currently, PDAC management is mainly based on radiological resectability criteria and biological criteria, such as CA 19.9 level (when available); definition of new preoperative prognostic markers, able to improve decision-making process (i.e. surgery upfront vs neoadjuvant therapy), is more and more necessary.

This is a good study with a consistent sample size; however, some issues should be addressed to improve the manuscript.

Question #1: Table 1 reported characteristics of study population: no data are reported about tumor grading, lymph nodal status, resection margin status (R0-R1), vascular resections, neoadjuvant regimens, adjuvant chemotherapy (which regimen? Administered or not?). These are well-known prognostic parameters that should be reported and investigated in order to define the real weight of CT attenuation on prognosis.

Author’s Comments: Thank you for your important comment. We describe the neoadjuvant chemotherapy regimen in the results as follows (page 4, line 140-141): “, and 19 (41.3%) patients with BRPC and 20 (20.4%) patients with LAPC underwent surgery after neoadjuvant treatment (folfirinox 34 [87.2%], gemcitabine 5 [12.8%]).” In addition, the surgical pathology results and adjuvant chemotherapy regimen are described as follows (page 5, line 172-179): “The postoperative pathology results of 198 patients who underwent surgical resection were as follows: tumor differentiation (well-differentiated 21 [10.6%], moderately-differentiated 160 [80.8%], poorly differentiated 17 [8.6%]), pT-stage (pT1 22 [11.1%], pT2 131 [66.2%], pT3 45 [22.7%]), metastatic lymph node status (pN0 81 [40.9%], pN1 78 [39.4%], pN2 39 [19.7%]), resection margin (R0 155 [78.3%], R1 43 [21.7%]), lymphovascular invasion (LVI- 78 [39.4%], LVI+ 120 [60.6%]). perineural invasion (PNI- 24 [12.1%], PNI+ 174 [87.9%]). Of the 198 patients who underwent surgical resection, 123 (62.1%) received adjuvant chemotherapy (gemcitabine 103 [52.0%], 5-fluorouracil 20 [10.1%]).”

Question #2: Preoperative CA 19.9 is a well-studied parameter and values > 500 kU/L are considered expression of aggressive tumor and defined as “biological” borderline-resectable disease, according to International Study Group of Pancreas Surgery (ISGPS). Authors should report at least mean or median value of analyzed subgroups and assess its prognostic weight, instead of categorization in “normal/elevated.” Decrease of CA 19.9 is considered a relevant response parameter to neoadjuvant chemotherapy: could authors add data about it in BR/LA PDAC subgroup?

Author’s Comments: Thank you for important comments. In Table 1, the mean ± standard deviation of CA19-9 is described as follows.

RPC (n = 159)

BRPC/LAPC (n = 139)

Total (n = 298)

p-value

CA 19-9 (U/mL)

495.2 ± 1,046.5

1,157.5 ± 2,783.0

804.1 ± 2,071.4

0.009

 The corresponding part of the result is also described as follows (page 4, line 149-151): “CA19-9 was significantly elevated in the BRPC/LAPC group compared to the RPC group (1,157.5 ± 2,783.0 vs. 495.2 ± 1,046.5 U/mL; p = 0.009).” In addition, the prognosis according to CA19-9 changes before and after cancer was described in the results as follows (page 8, line 243-247): “Among 139 patients in the BRPC/LAPC group, CA19-9 decreased in 93 patients (66.9%) and increased in 46 patients (33.1%) after neoadjuvant chemotherapy. Patients with decreased CA19-9 had a significantly longer survival period than those with elevated CA19-9 (18.6 months [95% CI = 15.1-22.1] vs. 11.2 months [95% CI = 10.1-12.4]; p < 0.001).”

Question #3: How were CT attenuation cut-off, reported in tab. 3 and 4, defined? Could pancreas texture affect tumour CT attenuation (i.e. “fatty pancreas, chronic pancreatitis, etc)? Please clarify.

Author’s Comments: Thank you for your comment. In this study, the cutoff value was selected using the X-tile program. In the Method part, it is described as follows (page 3, line 115-119): “Patients were divided into two groups using the X-tile program (Version 3.6.1, Yale University, New Haven, USA) [24], a bioinformatic tool used to determine optimal cut-off points for survival analysis. The X-tile software tested all possible cut-off points of the target quantitative data using log-rank tests and selected the lowest p-value.”. Patients with chronic pancreatitis or severe fatty pancreas were excluded from the study.

Question #4: In discussion section (lines 270-271): authors stated that prevalence of iso-attenuating PDAC lesions was higher in small or well-differentiated lesions but no data about grading are reported. Authors should add data about it in results section.

Author’s Comments: Thank you for your comment. The reference in this sentence is cited as follows (page 9, line 288-291): "In addition, the prevalence of iso-attenuating PDA lesions was significantly higher among patients with small (< 20 mm) or well-differentiated tumors than among those with moderately- or poorly-differentiated tumors [20]."

Question #5: Table S2: there is strong difference between pre- and post-neoadjuvant chemotherapy PVP attenuation in T1 BR/LA cancers. Is this difference significant? Could it be considered as better response of small lesions to chemotherapy? This is an interesting finding: authors should analyze it in discussion section.

Author’s Comments: Thank you for good comment. Paired t-test was performed to examine the difference in PVP (HU) values before and after chemotherapy by size group.

BRPC/LAPC (n = 139)

T1 (≤ 2 cm)

T2 (2-4 cm)

T3 (>4cm)

p-value

Initial CT

PVP (HU)

83.9 (52.8-114.9)

82.9 (77.5-88.3)

60.0 (54.7-65.2)

0.420

Follow-up CT

PVP (HU)

94.6 (72.9-116.3)

82.1 (76.7-87.4)

66.3 (59.6-73.1)

0.972

P-value (paired t-test)

0.404

0.803

0.071

Since there was no significant difference in PVP (HU) values in all of the T1, T2, and T3 groups, it was not described in the discussion.

The authors really appreciated the reviewer’s kind and accurate comments. The revision based on these comments made this manuscript more accurate and the quality improved. Thank you again.

Jin-Hyeok Hwang, M.D., Ph.D.

Dong Woo Shin, M.D.

Round 2

Reviewer 2 Report

Dear authors,

Thank you very much for reviewing, clarifying and modifying the allegations made previously. It is a very constructive work with high scientific rigor. I have communicated it to the editors of the journal.

Kind regards

Reviewer 3 Report

Thank you for the thorough revision. The manuscript has been significantly improved. I have no other issues.